# OpenReview forum: "Can We Trust Embodied Agents? Exploring Backdoor Attacks against Embodied LLM-Based Decision-Making Systems"
_ICLR.cc/2025/Conference — ICLR 2025 Poster_

### Official Review · Reviewer_eUW2 · 2024-10-25

**Soundness:** 3
**Presentation:** 3
**Contribution:** 2
**Rating:** 6
**Confidence:** 3

**Summary:**

This paper studies vulnerabilities of large language models (LLMs) with application scenarios in autonomous driving (examined via simulators) and robotics (examined on robots carrying out tasks in simulated home environments). The authors introduce backdoor attacks against LLM-based decision making systems (BALD), a setup that deploys attacks targeting different components of the LLM decision-making pipeline to activate malicious behaviors without degrading performance. BALD is evaluated using multiple LLMs on three different simulator environments.

*******
Updated my score after examining author response.
*******

**Strengths:**

1. Examining backdoor attacks focusing on embodied LLM systems is an open and important area of research at the intersection of AI and resilience.

2. Experiments are diverse in that multiple LLMs and distinct simulator environments are considered, which highlights broad applicability of the BALD framework.

3. DEmonstrable effectiveness of the three classes of backdoor attacks proposed, while performance on benign data samples is not affected.

**Weaknesses:**

1. I am not fully convinced by the thoroughness of the experimental evaluation. The only SOTA method BALD is compared with is BadChain. While performance of BALD is improved relative to BadChain, these results do not appear to provide a wider perspective into how well BALD compares to other SOTA mechanisms.

2. Absence of experimental evaluations on more recent LLMs (e.g., GPT-4).

3. Experiments focus on evaluation of BALD against inference-based defenses. It will be interesting to see how BALD performs against other defense mechanisms which might include adversarial training or dataset sanitization as part of the defense pipeline.

4. While the authors explicitly indicate that their experiments are carried out on simulated environments, it would have been interesting to see results on physical robotic systems in order to fully comprehend effectiveness of the proposed methodology.

Minor issues:
a. Line 508 - Typo: resilience --> resilient

b. Clarity of Fig. 1 can be improved, especially the distinction between word injection and knowledge injection can be made more clear in the figure or its caption.

**Questions:**

1. How does BALD compare to other SOTA methods beyond BadChain?

2. How is the performance of BALD on more recent LLMs, inlcuding Gemini and GPT-4?

3. How effective is BALD in bypassing different classes of defense mechanisms against backdoor attacks?

4. Is there a scope to evaluate BALD on a physical robotic system? If not, what challenges are anticipated (beyond procuring access to such a setup, if it is not available)?

---

> ### Author Response · Authors · 2024-11-19
> **Re: Reviewer-eUW2-1**
>
> Thanks for your feedback and question. Our responses are as follows:
>
> ***Weakness-1 & Question-1***
>
> As mentioned in the introduction and related work, To the best of our knowledge, we are the first to explore the full-channel backdoor attacks, and the first to consider the scenario-manipulation backdoor and the RAG-based backdoor for embodied LLM-based decision-making applications. We compare it with the recently published paper BadChain[r1]. However, even for BadChain, it doesn’t have exactly the same setting as ours, but we still try to include and compare with them (Even though BadChain is also called a backdoor attack, the poisoned examples are injected during in-context learning).
>
> Some other concurrent (not published as of our submission) works as we discussed in the related works, such as [r2], which only pure word-based attack against shopping web-agent)  and [r3], which attacks the knowledge base and retriever solely, have different settings and applications from ours.  We do discuss other related methods in related works but they are not comparable in terms of basic settings and assumptions.
>
> Therefore, in the AI security domain, we believe that there is no well-established SOTA because different attacks are highly tailored for different applications under specific assumptions.  We try to compare with some transferable and published attacks and compare our proposed different approaches to demonstrate the vulnerability of LLM-based embodied agents. We believe that given our main results, comparison to in-context backdoor attack and ablation studies, we have enough empirical evidence to support our main findings
>
> - The fine-tuning time backdoor attacks are more effective than in-contect learning backdoor attacks
> - The proposed scenario-based and RAG-based attacks are practical and stealthy attacks in the physical world
> - Different attacking mechanisms have their trade-offs in terms of effectiveness and stealthiness
> - We also discuss the corresponding defenses.
>
> ***Weakness-2 & Question-2***
>
> Thanks for the suggestion. Gemini focuses more on multi-modal tasks and GPT-4 didn’t open its API for fine-tuning until this September. During the rebuttal, as per your suggestion, we have been conducting experiments on the most recent GPT-4o models with the HighwayEnv data. We share a similar setting as the other experiments in the main table. The model is fine-tuned with 120 samples and tested on 100 samples under both benign scenarios and backdoor scenarios, respectively.  The primary results are shown as follows, which demonstrates the effectiveness of our methods. The results on  GPT-4o align with the ones on other LLM models. We will add the results on the other two datasets/simulations in the final version of the paper.
>
> | Metric                  | BALD-word | BALD-scene | BALD-RAG |
> |-------------------------|-----------|------------|----------|
> | Benign Accuracy (ACC)   | 100       | 98         | 90       |
> | Attack Success Rate (ASR) | 100       | 100        | 100      |
>
> ***Weakness-3 & Question-3***
>
> Thanks for your question. We agree it is important to understand how effective the BALD attacks are against different attacks. We conduct experiments with 5 different defenses, as mentioned in ***Reviewer-7z45-Weakness-1 and Question-7***. We indeed put more defenses in the Appendix given our threat model assumptions, as mentioned in RQ7 in Section 4.2.
>
>   - Inference-time defenses
>     - Benign in-context learning demonstrations (main paper).
>     - outlier rare word removal (main paper).
>   - Training-time defenses:
>     - random augmentation training (Appendix B).
>   - Post-training defense:
>     - Post-training benign fine-tuning (Appendix B).
>     - Adversarial training (Appendix B).
>
> To the best of our knowledge, we are the first to test whether benign in-context learning can defend the backdoor-attacked (misaligned) model, which we can consider as a reverse jailbreak inference. Interestingly, we find different models have very different performances: backdoor-attacked GPT3.5  is quite robust to such benign in-context examples while Palm2 isn’t. Besides, we conduct 3 kinds of fine-tuning time defense including data augmentation fine-tuning, benign post-fine-tuning, and adversarial fine-tuning, which is demonstrated in Appendix B in detail. Given the results there, we can conclude that it is still challenging to defend against the attacks without knowing the backdoor patterns in the first place.
>
> In terms of dataset sanitization, it's important to note that given our threat model assumption (Section 3.1), we assume the attacker takes control of the fine-tuning process and uploads the model. Therefore, dataset sanitization more applies to situations where attackers can only have partial access to the fine-tuning process and inject some poison data into the database. Therefore, dataset sanitization can be an important future research direction under this assumption.

---

> ### Author Response · Authors · 2024-11-19
> **Re: Reviewer-eUW2-2**
>
> ***Weakness-4 & Question-4***
>
> We highly appreciate this comment! We believe that our simulation-based results can be extended into real-world setups. We adopt the same simulation setups as in [r5, r6], where the same techniques are deployed in both the simulator and the real world. Therefore, by targeting their techniques and setups, which show practicability in the real world, we believe the simulator results should be persuasive enough to indicate real-world vulnerabilities.
>
> However, we also admitted that discussing the anticipated challenges in real-world attacks can further broaden our scope.
>
> - For BALD-word, as discussed in our main paper, it is challenging to inject words into real-world setups as it requires system intrusion, which modifies the input text during run-time. However, if voice-to-text conversion is used as the bridge for human-agent interaction (e.g. Siri in Apple device), the word trigger can be injected more easily.
>
> - For autonomous driving, although some works are demonstrating the conversion from LLM decision to low-level motion [r7, r8] and some companies are applying LLM/VLM to their autonomous driving systems, it is still challenging to integrate LLM/VLM into real driving systems[r9,r10]. All works as we listed in the related works use similar simulation settings.
>
> ---
> **Reference**
>
> [r1] Xiang, Zhen, et al. "Badchain: Backdoor chain-of-thought prompting for large language models.", ICLR 2024.
>
> [r2] Yang, Wenkai, et al. "Watch out for your agents! investigating backdoor threats to llm-based agents.", to be published in NeurIPS, 2024.
>
> [r3] Chen, Zhaorun, et al. "Agentpoison: Red-teaming llm agents via poisoning memory or knowledge bases.", to be published in NeurIPS, 2024.
>
> [r4] Qi, Fanchao, et al. "Onion: A simple and effective defense against textual backdoor attacks. EMNLP 2021
>
> [r5] Singh, Ishika, et al. "Progprompt: Generating situated robot task plans using large language models." ICRA 2023
>
> [r6] Zhang, Hangtao, et al. "BadRobot: Manipulating Embodied LLMs in the Physical World." arXiv preprint arXiv:2407.20242 (2024).
>
> [r7] Sha, Hao, et al. "Languagempc: Large language models as decision makers for autonomous driving." arXiv preprint arXiv:2310.03026 (2023).
>
> [r8] Wang, Yixuan, et al. "Empowering autonomous driving with large language models: A safety perspective." ICLR LLM Agents 2024
>
> [r9]https://medium.com/nuro/lambda-the-nuro-drivers-real-time-language-reasoning-model-7c3567b2d7b4
>
> [r10] https://waymo.com/blog/2024/10/ai-and-ml-at-waymo/

---

> > ### Comment · Reviewer_eUW2 · 2024-11-20
> > **Response to Author Comments**
> >
> > Dear Authors,
> >
> > Thank you for your detailed response to my questions. I acknowledge having read the response and have updated my score to reflect this.

---

> > > ### Author Response · Authors · 2024-11-26
> > >
> > > Thank you for your response and feedback! We believe that your insightful comments, along with our responses, have helped clarify and strengthen the contributions of the paper. Thanks again and we look forward to discussions if there are any further questions!

---

### Official Review · Reviewer_VnSU · 2024-10-27

**Soundness:** 3
**Presentation:** 3
**Contribution:** 2
**Rating:** 3
**Confidence:** 4

**Summary:**

The paper explores the risks of backdoor attacks in embodied AI systems like autonomous driving and robotics. The authors propose a framework named BALD to examine three distinct types of attacks: word injection, scenario manipulation, and knowledge injection.

**Strengths:**

The authors systematically identify and implement three distinct types of backdoor attacks, showing the breadth of vulnerabilities that LLM-based decision-making systems face.

The paper includes extensive experiments using popular LLMs like GPT-3.5, LLaMA2, and PaLM2 in multiple embodied AI scenarios, such as autonomous driving (HighwayEnv) and household robotics (VirtualHome).

**Weaknesses:**

The motivation and technical contribution of the paper are below my expectations of the ICLR papers.

From the AI aspect, the paper does not introduce new technical contributions. While the proposed attack strategies are systematically categorized, the novelty of the contributions is somewhat unclear. Existing literature on adversarial and backdoor attacks in AI systems shares similarities with the approach taken in this paper.

From the security perspective, the paper is targeting an imaginary threat model. While the paper presents compelling simulation-based results, it does not demonstrate real-world applicability or validation of the proposed attacks. In fact, embodied AI is still a "concept", and it is still impractical to let LLM operate your vehicle.

**Questions:**

The writing is clear, and I appreciate the efforts of the authors.

---

> ### Author Response · Authors · 2024-11-19
> **Re: Reviewer-VnSU-1**
>
> Thank you for your feedback and comments. We appreciate your recognition of the comprehensiveness of our proposed attack channels and experiments. To further clarify our work and highlight our contribution, we provide detailed responses below.
>
> ***Weakness-1***: *“From the AI aspect, the paper does not introduce new technical contributions. While the proposed attack strategies are systematically categorized, the novelty of the contributions is somewhat unclear. Existing literature on adversarial and backdoor attacks in AI systems shares similarities with the approach taken in this paper.”*
>
> In this work, we address three different types of backdoor attacks with varying data poisoning settings. We believe our approach makes multi-faceted contributions in comparison to existing literature, and we will further elaborate on them below.
>
> 1. **On the problem formulation:** We emphasize that our work, as outlined in the introduction, related works, and methodology sections, specifically targets novel backdoor mechanisms designed for new surfaces and challenges inherent to LLM-based embodied agents. These new surfaces and challenges include fine-tuning stage backdoor injection, broad usage of Retrieval-Augmented Generation (RAG), limited system access during run-time, and physical-world trigger scenarios. As discussed in related works, all backdoor attacks generally share a fundamental basis in data poisoning during training and triggering during inference. However, researchers are continually developing diverse attacking mechanisms, targeting various systems, and optimizing different facets of these attacks. Our approach addresses these challenges through a new framework tailored to LLM-based embodied agents. This framework introduces various attack channels and novel techniques, providing fresh insights into the vulnerabilities of such systems.
>
> 2. **On the attacking mechanisms:** We present three distinct attack mechanisms: word-based, scenario-based, and RAG-based, each targeting different surfaces with unique assumptions.
> - For the word injection mechanism, we developed a technique that disentangles trigger-word design from backdooring training, building on recent work [r1]. This straightforward but effective word-injection attack serves as a basis for the subsequent stealthier and more practical attack mechanisms.
> - In the scenario-based attack, we introduce a high-level semantic scenario or environment as triggers, diverging from traditional rare-word triggers. This approach addresses new challenges, such as mapping scenario triggers within physical environments (utilizing Scenic, a programmable scenario generator), enhancing scenario stealthiness and generalization (via LLM-based scenario augmentation), and preventing trigger leaks into benign scenarios (using novel contrastive sampling and reasoning methods).
> - For RAG-based attacks, our work is among the first to apply backdoor strategies to RAG-based LLMs. Unlike prior knowledge poisoning or simplistic trigger-word-based attacks, our approach integrates scenario-based retrieval triggers with word-based backdoor triggers to enhance stealth and efficacy.
> The above points represent significant technical contributions optimized for attacking LLM-based embodied AI systems.
>
> 3. **On the scientific findings and experimental insights:** We unveil several critical insights into the vulnerabilities of these systems, including:
> - Fine-tuning backdoor attacks can be highly effective with minimal data (fewer than 100 samples) and short training durations.
> - We systematically compare the strengths and weaknesses of different attack mechanisms (word, scenario, and knowledge-based).
> - Our work highlights the stealthiness of the proposed attacks in terms of both triggers and model behavior.
> - We explore both run-time and training-time defense mechanisms, demonstrating the robustness of our attacks. Additionally, we observed unique effects of in-context correct examples on backdoor-attacked models across various model families. Collectively, these findings reveal significant vulnerabilities in LLM-based decision-making pipelines in embodied AI and emphasize the urgent need for enhanced safety measures.

---

> ### Author Response · Authors · 2024-11-19
> **Re: Reviewer-VnSU-2**
>
> ***Weakness-2**: From the security perspective, the paper is targeting an imaginary threat model. While the paper presents compelling simulation-based results, it does not demonstrate real-world applicability or validation of the proposed attacks. In fact, embodied AI is still a "concept", and it is still impractical to let LLM operate your vehicle.*
>
> Thanks for your comments. We will address each of them below and explain the motivation of our work.
>
> 1. The first is about the **threat model**.
>
> According to the definition of Cisco [r22]:
>
> > Threat modeling is the process of using hypothetical scenarios, system diagrams, and testing to help secure systems and data. By identifying vulnerabilities, helping with risk assessment, and suggesting corrective action, threat modeling helps improve cybersecurity and trust in key business systems.
>
> Our threat model is based on reasonable hypothetical scenarios to analyze the vulnerabilities of the system, and follows most threat model assumptions in backdoor attacks, where the attacker injects the backdoor pattern in the training stage and triggers it during the inference stage. Moreover, beyond general LLM safety and backdoor attacks [r1, r2, r3, r4, r5], the work DriveLikeHuman [r6] DiLu [r7], and [r8] use LLM for decision-making for autonomous driving, the work of DiLu [r7] and [r9] utilize the RAG setting to extract similar and critical scenarios, and the works [r10] and Nuro company’s product ‘Lambda’ [r11] fine-tune or align pre-trained LLM/MLLM  for their autonomous driving or robotics tasks. All the attack surfaces we are considering in this paper, including fine-tuning, physical scenario, RAG, and language-based decision-making, have been explored for embodied AI systems and demonstrated their practical relevance. Therefore, we think our threat model follows the most prevalent assumptions of backdoor attacks and considers practically-relevant attack surfaces that are being actively studied in the community, and are not only "imaginary".
>
> 2. Secondly, regarding the **real-world applicability**.
>
> While we agree real-world experiments are often preferable, we believe that our simulation-based results are strong enough to support our findings and conclusions. For the robotics experiments, We adopt the same simulation setups as the ones in the previous related work [r12, r13] where the same techniques are deployed in both the simulator and the real world. Therefore, by targeting their techniques and setups, which have shown practicality in the real world, we believe the simulator results should be persuasive enough to indicate real-world vulnerabilities.
>
> In terms of autonomous driving experiments,  it is impractical for us to set up commercial autonomous vehicles operated by LLMs, as it's not mature enough as robotics. However, we follow the setting and apply attacks to the most commonly-used datasets and simulation platforms Highway Env (used by [r6, r7, r8]) and nuScenes/Carla (used by [r6, r14, r15]), which represent the most advanced achievements in this domain.
>
> 3. Thirdly, about our **motivation to attack embodied AI**.
>
> We believe security research should not only target mature systems, but also the vulnerabilities of emerging technologies, which is especially necessary before they are practically adopted. In this way, we can help developers realize potential risks and adequately address security vulnerabilities during the development stage and before the wide adoption. This is even more important as we consider safety-critical cyber-physical systems (cars, robots), where malfunctions under security attacks can lead to life-threatening consequences.
>
> In particular, embodied AI has drawn wide attention in both academia and industry, showing promising potential. For instance, (i) in robotics: research works like [r12, r16, r17, r18] have shown promising results of using LLMs for robot planning or even manipulation; and (ii) in autonomous driving: in academia, many research papers have been published recently, such as [r6, r7, r8, r19, r20], while in industry, autonomous companies have shown wide interests in integrating LLMs into their products (Nuro[r11], Waymo[r21]).
>
> ---
>
> As we provide these evidences above to highlight our motivation and approach, we recognize that there can be different perspectives on the urgency and importance of these potential security issues, and we look forward to further engaging in constructive discussions with you!

---

> ### Author Response · Authors · 2024-11-19
> **Re: Reviewer-VnSU-3**
>
> ---
> **Reference**
>
> [r1] Xiang, Zhen, et al. "Badchain: Backdoor chain-of-thought prompting for large language models.", ICLR 2024.
>
> [r2] Xu, Xilie, et al. "An LLM can Fool Itself: A Prompt-Based Adversarial Attack.", in ICLR, 2024
>
> [r3] Chen, Zhaorun, et al. "Agentpoison: Red-teaming llm agents via poisoning memory or knowledge bases.", NeurIPS, 2024.
>
> [r4] Yang, Wenkai, et al. "Watch out for your agents! investigating backdoor threats to llm-based agents.", NeurIPS, 2024.
>
> [r5] Zou, Wei, et al. “Poisonedrag: Knowledge Poisoning Attacks to Retrieval-Augmented Generation of Large Language Models.”, USENIX Security, 2024.
>
> [r6] Fu, Daocheng, et al. "Drive like a human: Rethinking autonomous driving with large language models." WACV 2024
>
> [r7] Wen, Licheng, et al. "Dilu: A knowledge-driven approach to autonomous driving with large language models. ICLR 2024
>
> [r8] Wang, Yixuan, et al. "Empowering autonomous driving with large language models: A safety perspective." ICLR LLM Agents 2024
>
> [r9] Han, Dongge, et al. "LLM-Personalize: Aligning LLM Planners with Human Preferences via Reinforced Self-Training for Housekeeping Robots.", MM, 2024
>
> [r10] Han, Dongge, et al. "LLM-Personalize: Aligning LLM Planners with Human Preferences via Reinforced Self-Training for Housekeeping Robots." arXiv preprint arXiv:2404.14285 (2024).
>
> [r11] https://medium.com/nuro/lambda-the-nuro-drivers-real-time-language-reasoning-model-7c3567b2d7b4
>
> [r12] Singh, Ishika, et al. "Progprompt: Generating situated robot task plans using large language models." ICRA 2023
>
> [r13] Zhang, Hangtao, et al. "BadRobot: Manipulating Embodied LLMs in the Physical World." arXiv preprint arXiv:2407.20242 (2024).
>
> [r14] Shao, Hao, et al. "Lmdrive: Closed-loop end-to-end driving with large language models." CVPR 2024.
>
> [r15] Sima, Chonghao, et al. "Drivelm: Driving with graph visual question answering. ECCV 2024
>
> [r16] Liang, Jacky, et al. "Code as policies: Language model programs for embodied control." ICRA 2023
>
> [r17] Song, Chan Hee, et al. "Llm-planner: Few-shot grounded planning for embodied agents with large language models." ICCV 2023
>
> [r18] Lin, Kevin, et al. "Text2motion: From natural language instructions to feasible plans." Autonomous Robots  2023
>
> [r19] Cui, Can, et al. "A survey on multimodal large language models for autonomous driving." WACV 2024
>
> [r20] Wei, Yuxi, et al. "Editable scene simulation for autonomous driving via collaborative llm-agents." CVPR 24
>
> [r21] https://waymo.com/blog/2024/10/ai-and-ml-at-waymo/
>
> [r22] https://www.cisco.com/c/en/us/products/security/what-is-threat-modeling.html

---

> > ### Comment · Reviewer_VnSU · 2024-11-22
> >
> > Thank you for your response. I do appreciate the merits of the work but your response did not solve my concerns. Some examples you have provided are not the same cases as my comments.

---

> > > ### Author Response · Authors · 2024-11-27
> > >
> > > Thank you for the response! Could you be more specific about which concerns have not been addressed and what examples you are referring to？ We look forward to engaging in an in-depth discussion with you on these concerns. Thanks!

---

### Official Review · Reviewer_CDE8 · 2024-10-29

**Soundness:** 4
**Presentation:** 3
**Contribution:** 4
**Rating:** 8
**Confidence:** 4

**Summary:**

This paper presents a comprehensive framework for Backdoor Attacks against LLM-based Decision-making systems (BALD) in embodied AI, identifying significant security vulnerabilities in such systems. The authors introduce three distinct attack mechanisms—word injection, scenario manipulation, and knowledge injection—targeting various components of LLM-based decision-making pipelines. Extensive experiments on representative LLMs in tasks like autonomous driving and home robotics demonstrate the effectiveness and stealth of these backdoor attacks. The results show near-perfect success rates for word and knowledge injection attacks, and high success rates for scenario manipulation without runtime system intrusion.

**Strengths:**

- I believe this paper provides a comprehensive exploration of decision-making vulnerabilities in embodied agents and proposes three forms of backdoor attack methods for evaluation.
- The proposed approach is novel and easy to understand. The authors present solid experimental results, evaluating across various LLMs and decision-making agents, as well as in different embodiedment scenarios.
- The paper is well-written, and the figures are clear and easy to comprehend. I greatly appreciate the discussion format in the experimental section of this paper. The questions raised by the authors and their responses provide valuable insights for the community.

**Weaknesses:**

Although it is still unclear what the upper limits of scenario manipulation attacks might be, I suggest that the authors consider more complex forms of visual backdoor injection, such as carefully designed patches, specific object combinations, or particular viewpoints. Currently, the backdoor injection primarily focuses on editing the text modality. I believe the authors could further explore the impact of complex visual modality injection on the results, as well as evaluate more advanced Large Multimodal Models (like GPT-4o) as decision agents. Of course, this is merely a suggestion, and the authors may choose to adopt it or provide alternative counterarguments.

**Questions:**

I believe this paper makes a significant contribution, providing valuable insights into the field of embodied intelligence security, and is deserving of acceptance at ICLR.

**Details Of Ethics Concerns:**

The backdoor methods proposed in this paper pose potential risks to real-world embodied applications. I recommend clearly indicating these risks and adding warning information in the text.

---

> ### Author Response · Authors · 2024-11-19
> **Re: Reviewer-CDE8**
>
> Thanks for your devoted time to reviewing and acknowledging our work. We highly appreciate your suggestions. As we also discussed in ***Reviewer 7z45 Question-3***, we mainly considering LLMs instead of VLMs since during the process of this work, many VLMs-based embodied agents have not been published yet. Additionally, how RAG should work in VLMs is still under research. We also mentioned that extending our work to VLMs in Appendix D.3 could be a very important future direction as these systems are getting more mature.

---

> ### Comment · Reviewer_CDE8 · 2024-11-28
>
> Thanks for your response. My concerns have been addressed so I decide to raise my confidence score.

---

> > ### Author Response · Authors · 2024-11-28
> >
> > Thank you for your valuable feedback and thoughtful comments! We are pleased that our response has addressed your concerns. Applying the proposed attack mechanisms, particularly BALD-scene and BALD-RAG, to VLM/MLLMs is indeed a promising direction. While it falls outside the scope of this work, we look forward to engaging in open discussions on potential new attack surfaces and mechanisms, which we believe will greatly benefit the embodied AI safety community. Thank you once again!

---

### Official Review · Reviewer_7z45 · 2024-11-04

**Soundness:** 3
**Presentation:** 3
**Contribution:** 3
**Rating:** 8
**Confidence:** 4

**Summary:**

This work presents a comprehensive framework for backdoor attacks, termed BALD, which targets LLM-based decision-making systems. The study explores three attack methods—word injection, scenario manipulation, and knowledge injection. The authors exploit different parts of the LLM decision-making pipeline. Experiments on models like GPT-3.5, LLaMA2, and PaLM2 in autonomous driving and home robot tasks reveal high success rates for these attacks, including nearly 100% for word and knowledge injection.

**Strengths:**

Novelty and Scope: The work is new. The paper is the very first to propose a framework, BALD, for studying backdoor vulnerabilities in LLM-based decision-making in embodied agents. The word injection, knowledge injection, and the scenario manipulation are comprehensive and provide valuable insights into safeguarding AI systems.

The evaluation of the experimentation is thorough. This paper depicts a wide experimentation on different platforms, such as autonomous driving: HighwayEnv, nuScenes; robotics: VirtualHome; and others, utilizing LLMs like GPT-3.5, LLaMA2, and PaLM2. Its claim and conclusion stand valid with the review.

Relevant for the real-world setting: The misguided action in an autonomous driving or robotics task poses potential real-world risks of a defective decision-making system and, therefore, is of high relevance for practitioners in AI safety.

The methodology section is very clear on the type of attack, with clear descriptions supported by diagrams, information on data preparation, and structured experiments. The structured nature makes the work more reproducible and allows other researchers to build upon this work.

**Weaknesses:**

Very few defenses are available. Although the paper does discuss some of the defenses concerning these types of attacks, such as benign demonstrations in in-context learning and ONION for word outlier removal, the defenses are relatively limited. What would have added more value to this work, especially for real-world deployment, was a stronger emphasis on countermeasures.

Lack of discussion of model-specific vulnerabilities: Whereas this study utilizes different variants of LLMs, namely GPT-3.5, LLaMA2, and PaLM2, shallow discussions of the nature of-or even how-vulnerability to backdoor attacks comes with specific model architectures are provided. For instance, the differences between open-source and proprietorial models or the differences in robustness from one model to another could offer a deeper insight.

Did not test for long-term effectiveness. The paper does not discuss the possibility of defenses or system self-adaptation over time that could reduce the efficacy of the attack. In real-world applications, the LLMs could conceivably learn or otherwise adapt defenses over continued use, thus reducing the long-term impact of backdoors-a possibility that would require the attacker to do continuous re-injection efforts.

The scenario is fine-grained: whether the attack with scenario manipulations succeeds depends on these fine-grained settings-for example, a red Mazda CX-5 with hazard lights in front. Without such fine-grained contextual descriptions, the manipulation of which would obviously lead to lesser variability, the reproduction in less controlled scenarios could be hard, hence leading to lesser reliability in the attack.

Lack of domain-generalization discussion. Although BALD shows promising results in the domains of autonomous driving and home robotics, there is a significant dearth of discussions regarding how these attacks would generalize across domains-such as healthcare and industrial automation. This paper would benefit from an explicit rendering of crossdomain applicability, specifically for high-stakes environments outside of robotics and driving.

**Questions:**

Thank you for submitting this work to ICLR! Overall, I like this paper as it is well-written and well-organized. While reading this work, I have the following comments and questions:

The authors proposed three backdoor attacks, BALD-word, BALD-scene, and BALD-RAG. In technical aspect, the BALD-word is a classical dirty-label backdoor; the BALD-scene can be treat as a clean-label backdoor, where the trigger is carefully designed as different scenario, and force the LLM agent use a benign COT to lead to malicious target; the BALD-RAG is poisoning the RAG system with scenario trigger and target pairs. Correct me if I am wrong.

Q1: From my perspective, the BALD-word is not new, as it follows the classical setting, just pair the trigger with malicious output; many existing work has done that, for example[1,2,3], that the attacker has to be able to change the user's prompt during inference. Can the author clarify what is the major difference of BALD-word in poisoning agent with poisoning LLM?

Q2: For the BALD-scene attack, I personally like this attack format, as the attacker can attack the agent by changing the environment. However, I remember there are multiple papers that crafting adversarial examples to affect the autonomous driving system (e.g., 4-5), what prevent the authors from using adversarial attack to attack the LLM agent?

Q3: Same in BALD-scene attack, the authors feed images to VLM to generate text, and then encode text trigger into the text to form the poison dataset; I doubt if this is a realistic setting because there are many multi-modal LLMs who take image as input and directly produce output, which means there is no place to insert the text trigger. Does the author consider this scenario?

Q4: For BALD-RAG attack, I didn't get how its design make it outperform the existing work, from my understanding, it is a BALD-word attack and replacing the trigger word as scenario description, which are not very novel. Compared with the recent work [7], I didn't find technical contribution in the RAG setting.

Q5: In table 2, the FAR for PALM2 is higher than the other, can the author explain what is the possible reason for it?

Q6: In page 9, the author said "In contrast, we employed a more general scenario (i.e., a gray trash bin) on nuScenes dataset. This results in a significantly lower retrieval success rate from the knowledge database. However, it consistently triggers the backdoor attack
with an ASR of 100% when the poisoned knowledge is retrieved." I am not sure the logic is right, while the nuScenes result in low retrieval success rate, but get 100% ASR. How do you calculate ASR, should it be (retrieval rate) * (ASR after retrieve)?

Q7: The author only test common defenses such as outlier word removal, fine-tuning, which are pretty old defense (the newest one test is 2021). To prove the robustness of this attack, I encourage the author try more newer backdoor defenses.

[1] Spinning language models: Risks of propaganda-as-a-service and countermeasures.
[2] Poisoning language models during instruction tuning
[3] Removing rlhf protections in gpt-4 via fine-tuning
[4] Poltergeist: Acoustic adversarial machine learning against cameras and computer vision
[5] Evaluating the robustness of semantic segmentation for autonomous driving against real-world adversarial patch attacks
[7] AgentPoison: Red-teaming LLM Agents via Poisoning Memory or Knowledge Bases

---

> ### Author Response · Authors · 2024-11-19
> **Re: Reviewer-7z45-1**
>
> We would like to thank you for the insightful and very detailed comments! Our responses are as follows:
>
> ***Weakness-1 and Question-7***:
>
> Thank you for your valuable feedback. In this work, our primary contribution lies on the attack side, which forms the central focus and evaluation of the paper. However, we have also explored a range of defenses spanning different stages of the lifecycle of LLMs. Specifically, we consider five distinct defense setups:
>   - Inference-time defenses
>     - Benign in-context learning demonstrations (main paper).
>     - outlier rare word removal (main paper).
>   - Training-time defenses:
>     - random augmentation training (Appendix B).
>   - Post-training defense:
>     - Post-training benign fine-tuning (Appendix B).
>     - Adversarial training (Appendix B).
>
> These defenses were selected to provide coverage across key phases in the lifecycle of embodied LLMs: **fine-tuning** (training-time defenses), **evolving** (safety alignment,  post-training defenses), and **deployment** (inference-time defenses).  For the in-context learning defense, we are the first to explore this battle between misalignment and benign in-context learning. We observe interesting phenomena in which different attacked models perform differently when fed enough benign in-context examples, a novel aspect of understanding the model’s vulnerability.
>
> We appreciate the suggestion to emphasize countermeasures further. To this end, we combined two inference-time defenses and presented the results below for your reference:
>
> | Defense            | BALD-word ASR | BALD-scene ASR |
> |--------------------|---------------|----------------|
> | Without defense    | 100.0         | 78.0           |
> | ONION (top_k=10)   | 24.0          | 76.0           |
> | Benign ICL (k=3)   | 94.0          | 82.0           |
> | Combined               | 20.0          | 70.0           |
>
> ***Weakness-2***:
>
> Thank you for highlighting the limited discussion on model-specific vulnerabilities. The primary focus of our work is to demonstrate the general vulnerabilities present in the fine-tuning stage of embodied LLMs in decision-making tasks. To establish the generality of our attack frameworks, we evaluated three representative LLMs: GPT-3.5, LLaMA2, and PaLM-2.
>
> While we acknowledge the importance of model-specific vulnerabilities, several factors make a rigorous comparison challenging. For instance, we have full control over the fine-tuning parameters and techniques (e.g., LoRA) for open-source models like LLaMA. In contrast, for proprietary models like GPT-3.5 and PaLM-2, we are restricted to black-box access through fine-tuning APIs, limiting our ability to comprehensively analyze and compare the effects of fine-tuning configurations.
>
> However, we still observe some variations among different models. For instance, in the context of BALD-scenario attacks (physical scenario manipulation), we find it more challenging to inject scenario triggers into PaLM-2 compared to LLaMA, as the Attack Success Rate (ASR) and False Alarm Rate (FAR) for PaLM-2 are significantly lower. This may be attributed to the inherent robustness of the PaLM-2 model or potentially to Google's black-box fine-tuning techniques, which could mitigate fundamental changes in the model's behavior. In our in-context learning defense study, as illustrated in Figure 7c, we observe that certain models, such as PaLM-2, can correct their responses when presented with benign in-context examples. In contrast, others, like GPT-3.5, persist in producing backdoor-affected answers. These findings highlight that different models do not respond identically to attacks and defenses. However, our primary focus remains to demonstrate that the proposed attacks are broadly effective across various model families.
>
> ***Weakness-3***:
>
> Thank you for your insightful suggestions. We agree that system defenses and self-adaptation over time could affect the long-term effectiveness of backdoor attacks. To address this, we consider two scenarios:
>
> - **Partial Control of Model Development**:
>   If the attacker can only poison fine-tuning data, the longevity of backdoor triggers is crucial. As detailed in Appendix B (“Benign Post-Fine-Tuning”), fine-tuning on benign data after backdoor injection partially mitigates triggers. This raises two future directions: (i) designing backdoor triggers resilient to post-fine-tuning, and (ii) detecting/removing backdoors after injection.
>
> - **Full Control of Model Development**:
>   When the attacker has full control (Sec 3.1), such as maliciously releasing models on platforms like HuggingFace or via insider threats, they can continually inject triggers into cutting-edge models, regardless of the model evolvement.

---

> ### Author Response · Authors · 2024-11-19
> **Re: Reviewer-7z45-2**
>
> ***Weakness-4***:
>
> The fine-grained scenario arises from the black-box assumption of the RAG retriever, which is strict but realistic. RAG systems in decision-making retrieve similar scenarios as references, and under black-box settings, attackers maximize retrieval rates by ensuring high similarity between the current scenario and poisoned database entries. For instance, manipulating a scenario (e.g., adding a car with hazard lights) is more practical than directly injecting text into the system.
>
> If attackers have white-box access to the RAG process, more sophisticated attacks could bypass the need for fine-grained scenarios, achieving high retrieval rates through targeted design. The attack’s strength scales with the attacker’s knowledge; greater system access enables more powerful and reliable strategies. Investigating white-box attacks across representative RAG mechanisms is a promising but complex direction, which can be an individual future study alone.
>
> ***Weakness-5***:
>
> Thanks for your comments. Our scope mainly focuses on the decision-making of embodied AI systems. Our assumption of the LLM-based embodied AI systems and threat models is not limited to vehicles and robots. The attack can be generalized to other systems that have such word/language-based interfaces, operate in the physical world, or retrieve knowledge/experience by RAG, as long as their decision-making making is powered by LLMs. Autonomous driving and robotics are two of the most representative and safety-critical tasks of embodied AI, as shown by the increasing attention from both academia and industry.  Therefore, we believe our attack results on both tasks with three different settings (highway decision-making, physical world driving reasoning, home robot action generation) already show the generality of our attacks. However, we also consider healthcare and industrial automation as important scenarios, which could be an interesting direction for future research.
>
> ***Question-1***:
>
> Indeed, BALD-word is the most straightforward backdoor attack, sharing similarities in setting and methods with prior work. However, the novelty and necessity of studying this approach lie in two key aspects:
>
> 1. **Methodology**:
>    - We attack the CoT-based decision-making pipeline, distinct from traditional backdoor attacks targeting binary classification tasks (e.g., sentiment classification [1,2,3]).
>    - Existing works require backdoor training/fine-tuning with trigger words, which is computationally expensive for LLMs. Instead, we leverage recent theoretical results ([r1,r2] few-shot in-context learning approximates fine-tuning) and in-context backdoor methods ([r3]) to separate trigger optimization (via ICL) from backdoor fine-tuning.
>    - Our experiments reveal rare words outperform contextually connected words/phrases for backdoor attacks. We provide a comparison in the table below.
>
> | Trigger Type                   | ASR   | ACC   |
> |--------------------------------|-------|-------|
> | Rare words/phrases             | 100   | 100   |
> | Words/phrases with contextual connection | 84.0  | 96.0  |
>
> 2. **Completeness**:
>    - To cover all attack channels for embodied systems, studying word-based attacks is essential for a complete analysis.
>
> In summary, while we introduce a new optimization method and apply it to embodied AI decision-making tasks, we don’t want to overclaim the novelty of word injection. We focus on its design and trade-offs in novel decision-making tasks to ensure completeness. This foundation supports our more advanced scenario manipulation and RAG-based knowledge poisoning attacks.
>
> ***Question-2***:
>
> Thanks for your question and your acknowledgment of our attacks. The paper you mentioned differs from ours in the following aspects:
> - they focus on task-specific models, which are not LLMs-powered embodied AI in our context.
> - they are both attacking the perception module, but ours focuses on the decision-making (i.e., planning) module.
>
> Therefore, the techniques fundamentally differ from each other. Our main idea in scenario manipulation is inspired by model alignment, we believe that if LLMs can be aligned to human morality, there are possibilities to misalign the model to perform malicious behavior under certain conditions.
>
> ***Question-3***:
>
> Thanks for your careful consideration! Yes, we mainly considering LLMs instead of VLMs since during the process of this work, many VLMs-based embodied agents have not been published yet. Additionally, how RAG should work in VLMs is still under research. We also mentioned that extending our work to VLMs in Appendix D.3 could be a critical future direction as these systems are getting more mature.

---

> ### Author Response · Authors · 2024-11-19
> **Re: Reviewer-7z45-3**
>
> ***Question-4***:
>
> For the BALD-RAG attack, we don’t replace the trigger word with a trigger scenario, but we use both scenario-trigger and word-trigger together to disentangle the retrieval and backdoor triggering and increase the stealthiness of the attack. We have a different threat model from the concurrent work [7]. Specifically, there are two key points of the BALD-RAG design:
>
> 1. the scenario in the RAG database is only used for retrieval, and the model is not backdoor-trained with a scenario trigger (because in BALD-scene, we find it difficult to directly train a backdoor model with a scenario trigger).
>
> 2. The model itself is backdoor-trained with word-based triggers in the in-context examples. The poisoned knowledge is factually correct but only contains the trigger word.
>
> In this way, we disentangle the backdoor training and the optimization for retrieval. We can continually change/optimize the trigger scenario without the need to retrain the backdoor model. Also, the poisoned knowledge itself is correct, which increases the stealthiness. For the work [7], their threat model is to solely optimize the retrieval model to retrieve knowledge that contains misleading information, and they don’t conduct any fine-tuning with backdoor triggers.
>
> ***Question-5***:
>
> Thanks for the detailed observation: that’s also a finding that interests us during experiments. Although the PaLM2 fine-tuning is based on their API and we don’t have full control, we would love to share our insights on this. Related to your previous comments on model-specific vulnerability, we notice that it is more difficult to conduct scenario-manipulation attacks against PaLM2 model compared to the LLaMA and GPT-3.5 with similar settings - either the false alarm rate is high (in HighwayEnv data) or the attack success is low (in nuScene+Carla). As discussed previously in W2 and demonstrated in Figure 7c, we believe these results reveal that PaLM 2 is more robust to certain backdoor attacks.
>
> ***Question-6***
>
> Thanks for your careful review. Yes, for use * to mark the results in Table 2, the results of * represent ASR when the trigger is successfully retrieved. We will make this part more clear in the revised version. The end-to-end attack success rate of BALD-RAG on nuScenes-Carla is 35.5%. Compared to the one on HighwayEnv (ASR of 95.2), we want to emphasize the importance of retrieval rate by crafting fine-grained scenarios.
>
> ---
> **Reference**
>
> [r1] Dai, Damai, et al. "Why can gpt learn in-context? language models implicitly perform gradient descent as meta-optimizers. ACL 2023
>
> [r2] Irie, Kazuki, Róbert Csordás, and Jürgen Schmidhuber. "The dual form of neural networks revisited: Connecting test time predictions to training patterns via spotlights of attention." ICML 2022
>
> [r3] Xiang, Zhen, et al. "Badchain: Backdoor chain-of-thought prompting for large language models.", ICLR 2024.

---

> > ### Comment · Reviewer_7z45 · 2024-12-02
> >
> > Thanks for the detailed answer, I will raise my score accordingly.

---

### Meta-Review · Area_Chair_WG3P · 2024-12-20

**Metareview:**

The work studied the backdoor attack against embodied LLM-based decision-making systems. It considers three distinct attack mechanisms (word injection, scenario manipulation, and knowledge injection), and evaluates on several platforms.

It received 4 detailed reviews. Most reviewers recognized the strengths including the importance of the first backdoor attack against embodied AI, the clear writing, extensive evaluations on several LLM and platforms.

Meanwhile, there are also important concerns, including the lack of physical evaluations on real embodied AI systems, the lack of technical novelties compared to backdoor attacks against other AI systems. These points are indeed important limitations of this work. However, considering the potential importance of this new task, these limitations are acceptable.

After rebuttal and discussions, three reviewers raised their scores, while 1 reviewer keeps the score 3, as the reviewer thought that some concerns are not well understood.

Overall, I think the contributions of this work outweigh its limitations, thus my recommendation is accept. However, it is strongly required that the additional experiments and analysis, as well as the discussion about its limitations, will be added into the final version of the manuscript.

**Additional Comments On Reviewer Discussion:**

The rebuttal and discussions, as well as their influences in the decision, have been summarized in the above metareview.

---

### Decision · Program_Chairs · 2025-01-22

Accept (Poster)